# Infection-Induced Rhabdomyolysis in a Pregnant Woman with Undiagnosed Myotonic Dystrophy: A Case Report

**DOI:** 10.3390/medicina59050824

**Published:** 2023-04-23

**Authors:** Hyun Mi Kim, Heejeong Kim, Hyun-Hwa Cha, Haemin Kim, Hyo-Shin Kim, Mi Ju Kim

**Affiliations:** 1Department of Obstetrics and Gynecology, Kyungpook National University Hospital, Kyungpook National University School of Medicine, Daegu 41944, Republic of Korea; 2Department of Obstetrics and Gynecology, Kyungpook National University Chilgok Hospital, Kyungpook National University School of Medicine, Daegu 41404, Republic of Korea

**Keywords:** acute pyelonephritis, myotonic dystrophy, polyhydramnios, pregnancy, rhabdomyolysis

## Abstract

A 34-year-old nulliparous gravid female presented with acute bilateral pyelonephritis at 29 + 5 weeks gestation. The patient was relatively well until two weeks ago when a slight increase in amniotic fluid was noted. Further investigation revealed myoglobinuria and significantly elevated levels of creatine phosphokinase. The patient was subsequently diagnosed with rhabdomyolysis. Twelve hours after admission, the patient noted reduced fetal movements. A non-stress test revealed fetal bradycardia and non-reassuring variability in fetal heart rate. An emergency cesarean section was performed, and a “floppy” female child was delivered. Genetic testing revealed congenital myotonic dystrophy, and the mother was also diagnosed with myotonic dystrophy. Rhabdomyolysis has a very low incidence in pregnancy. Herein, we report a rare case of myotonic dystrophy with rhabdomyolysis in a gravid female with no history of myotonic dystrophy. Acute pyelonephritis is a causative agent of rhabdomyolysis that results in preterm birth.

## 1. Introduction

Rhabdomyolysis is a potentially life-threatening condition caused by the rapid dissolution of damaged or injured skeletal muscle fibers. It is characterized by a marked elevation in serum creatine kinase (CK) levels [1]. The incidence of rhabdomyolysis is rare in pregnancy. Probable causes include the use of ritodrine as a tocolytic, prolonged labor, extended immobility, hypokalemia, and myotonic dystrophy in patients with underlying disease [1]. Myotonic dystrophy, although its prevalence is low in Asian populations, is reported to be associated with rhabdomyolysis. Since rhabdomyolysis itself is rare in pregnancy, when it happens, it is necessary to evaluate whether the patient has typical features of myotonic dystrophy in a retrograde fashion.

Herein, we report a rare case of rhabdomyolysis secondary to a urinary tract infection in a gravid female with undiagnosed myotonic dystrophy.

## 2. Case Study

A 34-year-old nulliparous gravid female visited the emergency room with a one-day history of pyrexia (>39 °C) at 29 + 5 weeks gestation. On presentation, the patient’s vital signs were as follows: blood pressure, 140/72 mmHg; heart rate, 114 bpm; and body temperature, 39.4 °C. Physical examination revealed that the patient was febrile, experienced chills, and had myalgia and mild flank pain. There were no respiratory symptoms such as coughing or sore throat, suggesting coronavirus disease 2019 was absent. Other pyretic conditions, such as influenza, were ruled out.

The patient’s height was 157 cm. Her pre-pregnancy weight was 39 kg (body mass index [BMI]:16.4), while her weight on admission was 54 kg (BMI: 22.8). The patient was small and thin, and she had no known significant medical, surgical, or family history. The patient had a normal pregnancy without significant findings until two weeks prior to admission when a mildly increased amniotic fluid index (AFI) was noted during a routine prenatal checkup (Figure 1A). Her liver enzymes also showed slightly elevated values compared with that at baseline at the beginning of pregnancy.

Her complete blood count revealed leukocytosis, 15,270 μL. Further investigations showed an elevated C-reactive protein (CRP) level, 1.56 mg/dL, and elevated liver enzymes (aspartate aminotransferase/alanine aminotransferase (AST/ALT): 179/94 U/L). Her urine was dark brown, and urinalysis revealed hematuria, proteinuria, and pyuria. Blood and urine cultures were performed, and her urine culture was positive for *Escherichia coli* five days later. Renal ultrasonography revealed bilateral hydronephrosis, increased parenchymal echogenicity in both kidneys, indicating a high suspicion of acute pyelonephritis (APN). On fetal ultrasound, fetal heart rate was regular at 150–170 bpm, and doppler of the umbilical artery and middle cerebral artery, indicating the condition of the fetus, showed no specific findings. However, the AFI was 39, indicating polyhydramnios (Figure 1B).

A non-stress test (NST) revealed reactive fetal heart rate variability with regular uterine contractions (Figure 2A). Therefore, the patient was diagnosed with APN accompanied by preterm labor. Her management included antibiotics and tocolytics. Primary choice of tocolytics for her, since she had APN, was atosiban instead of ritodrine to prevent respiratory insufficiency caused by endotoxin-induced permeability edema. As the gestational age was earlier than 34 weeks, antenatal corticosteroids were also administered to enhance fetal lung maturation. Non-steroidal anti-inflammatory drugs were administered with fluid therapy as part of her conservative management. However, 12 h after admission, her leukocytosis worsened (24,410 μL) despite antibiotic and anti-inflammatory treatment, and repeated urinalysis revealed myoglobin levels exceeding 20,000 ng/mL with dark brown urine. We began to suspect rhabdomyolysis after her urine analysis showed prominent myoglobinuria. We thus began to perform additional blood tests to either diagnose or rule out rhabdomyolysis. Her blood tests revealed notably elevated levels of creatine phosphokinase (21,962 U/L), myoglobin (5381.62 ng/mL), creatine phosphokinase-MB fraction (48.09 ng/mL), and lactate dehydrogenase (831 U/L). Rhabdomyolysis was confirmed, and massive fluid therapy was initiated. Despite these treatments, however, the patient complained of reduced fetal movements. Repeated NST showed declining fetal heart rates (Figure 2B,C). Fetal bradycardia gradually aggravated on the NST (Figure 2D). Despite oxygen therapy and being in a lateral decubitus position, the fetal heart rate gradually fell to ˂100 bpm with minimal fetal variability. At 29 + 6 weeks of gestation, an emergency cesarean section was performed under spinal anesthesia. A female neonate was delivered, weighing 1410 g, without respiratory effort or fetal movements. The neonate was diffusely cyanosed and presented with Apgar scores of 1 at 1 min and 2 at 5 min. The pH of umbilical artery was 7.307, and base excess was measured at −3.9. The neonate was immediately intubated and admitted to the neonatal intensive care unit (NICU). Even so, she had a cardiac arrest on day 7 of life and developed severe intracranial hemorrhage (grade 4). Genetic testing performed on day 3 of life for suspected myotonic dystrophy owing to the absence of tone revealed that the CTG repeat sequence of the dystrophia myotonica protein kinase (DMPK) gene was increased by ˃150 times. Following the diagnosis of myotonic dystrophy in the newborn, the mother underwent a DMPK genetic test. The test confirmed that the mother had the same mutation. Maternal placental biopsy failed to reveal findings suggestive of chorioamnionitis. Six days after delivery, antibiotic treatment was completed, symptoms and laboratory test results improved, and the mother was discharged.

## 3. Discussion

Kidney complication is often associated with pregnancy, especially in the second trimester when the uterus begins to enlarge. Also, one of the frequent medical complications during pregnancy is APN. About 1–2% of pregnant women experience this condition which can lead to significant harm to both the mother and the fetus. Nulliparity and young age are risk factors. In over 50% of cases, pyelonephritis affects only one kidney, usually right-side, while about 25% of cases involve both kidneys. The enlarged uterus compresses the ureter, causing obstruction and stasis, which causes secondary APN [2]. Other etiologic factors for bilateral ureteral obstruction that might lead to not only APN but also acute renal insufficiency include polyhydramnios and multi-gestational pregnancies [3]. During pregnancy, pyelonephritis was the primary cause of septic shock. Additionally, urosepsis in the pregnant woman can increase the likelihood of preterm newborns developing cerebral palsy [4].

While there are well-known etiologies of either hydronephrosis or APN (or any kidney complications of pregnancy), bilateral kidney complications are uncommon and therefore need close attention. Evaluation of the patient’s history or meticulous physical examination may be helpful to rule out some rare causes of bilateral kidney complications.

In this case, severe polyhydramnios is believed to have been the causative agent for bilateral urinary tract obstruction, observed in a gravid female of small stature, that culminated in bilateral hydronephrosis and APN. It is also believed that a urinary tract infection in the same patient triggered rhabdomyolysis.

Rhabdomyolysis is a serious medical condition that can be life-threatening in adults, not to mention pregnant women. Acute myofiber necrosis, characterized by elevated serum CK levels, with or without myoglobinuria, causes rhabdomyolysis. Clinically, rhabdomyolysis presents with three distinct symptoms: myalgia, weakness, and myoglobinuria. Classically, myoglobinuria manifests as tea-colored urine. However, discolored urine may be the only presenting complaint of over 50% of the patients. Less than 10% of patients present with all three symptoms. Therefore, accurately diagnosing rhabdomyolysis with symptoms is difficult [5]. Several factors, including acquired and genetic factors, may induce rhabdomyolysis. Common causes can lead to rhabdomyolysis, such as severe exercises, crush injuries, burns, falls, immobilization for an extended period, tetanus, status epilepticus, drugs, malignant hyperthermia, hypothyroidism, and infection. If a healthy individual experiences a single episode of rhabdomyolysis, further diagnostic testing may not be necessary if there is an obvious and identifiable cause. On the other hand, the presence of a recurring history of rhabdomyolysis and pre-existing conditions, such as exercise intolerance, exercise-induced muscle cramps, muscle weakness, or a family history of myopathy, raise concerns about the possibility of an inherited muscle disorder. Rhabdomyolysis due to myopathy can be difficult to diagnose, especially in the absence of the aforementioned history. Especially, this could potentially be related to an obstetric emergency or comorbidity complicating the antepartum or peripartum period, and the obstetricians should be aware [6].

In this case, infection was a likely cause of rhabdomyolysis and an aggravating factor at the same time. Her infection source would be a urinary tract infection, and when it progressed to APN, it caused hyperthermia. Her background myotonic dystrophic condition is also more susceptible to rhabdomyolysis when she experienced fever.

Rhabdomyolysis is closely related to kidney failure due to the direct effect of myoglobin. A Canadian study showed that pregnant women who experienced AKI had more incidents of preeclampsia, obstetrical hemorrhage, and placental abruption [7].

In this case, infection was initially thought to have caused the rhabdomyolysis because this was the first episode of rhabdomyolysis experienced by a previously healthy patient. Furthermore, myotonic dystrophy was initially not suspected since the patient was unaware of its clinical characteristics prior to delivery and did not have a positive family history for it.

However, myotonic dystrophy should have been suspected in the following situations. First, it should be suspected in cases where hepatic enzymes are elevated, with unknown etiologies, several months before admission. While the pathophysiology of myotonic dystrophy is yet to be fully elucidated, a previous case report documented elevated liver enzymes without indications of chronic liver disease months before a diagnosis of myotonic dystrophy [8]. Second, it should be suspected in cases where polyhydramnios is diagnosed after a rapid increase in amniotic fluid over two weeks. It should also be considered in cases where the NST shows decreased fetal movement and bradycardia despite treatment during hospitalization. According to previous reports, congenital myotonic dystrophy of the fetus can be suspected if severe polyhydramnios, talipes, and decreased fetal movement are present; in these cases, additional testing is recommended [9,10]. Third, in cases involving APNs, progression to rhabdomyolysis during pregnancy is rare. Rhabdomyolysis during pregnancy is also rare. An incidence of infection-induced rhabdomyolysis during pregnancy is yet to be reported in the literature. Immobility due to prolonged labor, drugs including ritodrine, cocaine, and baking soda, and underlying diseases such as severe hypokalemia secondary to distal tubular acidosis and muscular dystrophy have been previously reported [1,6,11,12,13,14]. Finally, maternal rhabdomyolysis dramatically improves immediately after delivery in the absence of massive fluid administration. This postpartum recovery results from the redistribution of body fluids after decompression following delivery.

Despite conservative treatment, the condition of the fetus rapidly deteriorated, which is thought to be caused by bacterial endotoxins in APN. The fetus with congenital myotonic dystrophy may be more susceptible to endotoxin than a normal fetus. Endotoxins are known to cause fetal distress in animal studies. In this study, it was observed that giving *Escherichia coli* endotoxin to a pregnant baboon near term enhances the activity of the uterus, which can result in fetal distress and, ultimately, intrauterine death. The cause of fetal death in these cases is likely due to progressively severe hypotension in the pregnant baboon, which leads to reduced blood flow in the uterus, less oxygen supply to the fetus, and metabolic acidosis in both mother and the fetus. A decline in fetal arterial oxygenation suggests a reduction in the perfusion of intervillous space [15]. Accordingly, bacterial endotoxin-induced APN may lead to not only these conditions but also the rapid deterioration of fetal congenital myotonic dystrophy symptoms. In the presence of the endotoxin, the muscle tone of the fetus progressively decreased, including that of the cardiovascular and respiratory systems. The mother may have detected this decrease in fetal muscle tone as reduced fetal movement. The fetus exhibited congenital myotonic dystrophy with severe symptoms, despite the mother having asymptomatic myotonic dystrophy. Therefore, the fetal response to the endotoxins was rapid.

Myotonic dystrophy, especially type 1 myotonic dystrophy (DM1), is an autosomal dominant genetic disorder that results from an expansion of the CTG trinucleotide repeat on the DMPK gene on chromosome 19. The severity of the disease increases with successive generations, and patients become symptomatic at earlier ages. This is related to the size of abnormal CTG repeats, which expands in parent-to-child transmission, which is called anticipation. Symptoms also aggravate as generation proceeds. The clinical manifestations of DM1 include progressive muscular weakness, myotonia, which is the inability of muscles to relax after being contracted, cataracts, and endocrine abnormalities. Congenital myotonic dystrophy is the most severe form of myotonic dystrophy, which is characterized by generalized hypotonia, respiratory failure, and a high incidence of death in neonates or infants with severe cases. Infants with this condition may display several clinical features, including weakness in the facial muscles, a “fish-shaped mouth”, weak crying, and difficulty with feeding due to poor sucking. The mortality rate in neonates with this condition is relatively high, and most of them require ICU management after birth. Women with myotonic dystrophy have a significantly increased risk for adverse pregnancy outcomes because the marked aggravation of muscle weakness occurs during pregnancy. In particular, they have an increased risk of preterm labor, preterm premature rupture of membranes, preeclampsia, venous thromboembolism, and cardiac dysrhythmia, requiring blood transfusions and delivering via cesarean sections [9,16].

Patients with myotonic dystrophy are likely to develop rhabdomyolysis. The pathophysiology of this phenomenon is unknown. Even so, rhabdomyolysis might be the manifestation of a combination of environmental factors and predisposing genotypes. For example, environmental factors such as exercise, fever, infection, and drugs can induce rhabdomyolysis in susceptible patients such as muscular dystrophy [17].

However, as observed in the present case, the prompt diagnosis of muscular dystrophy in emergencies, such as emergency cesarean sections, is difficult. Obstetricians and anesthesiologists should be especially careful when administering medication to patients suspected of myotonic dystrophy. Anesthetic drugs, such as sedatives, would have severely aggravated the myotonic dystrophic condition, even to a fatal extent. Also, a multi-disciplinary approach with the department of pediatrics is required in such cases since congenital myotonic dystrophy correlates with significant perinatal morbidity and mortality from severe respiratory suppression. The active involvement of experienced pediatricians is required at delivery when congenital myotonic dystrophy is suspected. Additionally, newborns with features of myotonic dystrophy should immediately be transported to a NICU equipped with ventilatory support after delivery.

## 4. Conclusions

In summary, herein, a rare case of pregnancy complicated by congenital myotonic dystrophy, and rhabdomyolysis was reported. Bilateral hydronephrosis and accompanied pyelonephritis of both kidneys were observed, while at this point, either myotonic dystrophy or rhabdomyolysis was not suspected. Features such as polyhydramnios, bilateral kidney dysfunction, and rhabdomyolysis, which are all quite uncommon during pregnancy, may have been hidden clues of rare underlying diseases of pregnant women. As myotonic dystrophy may easily be misdiagnosed in the absence of other distinguishing features, muscle-related symptoms, or a family history of muscle disease, this case is clinically relevant for obstetricians, anesthesiologists, and pediatricians.

## Figures and Tables

**Figure 1 medicina-59-00824-f001:**
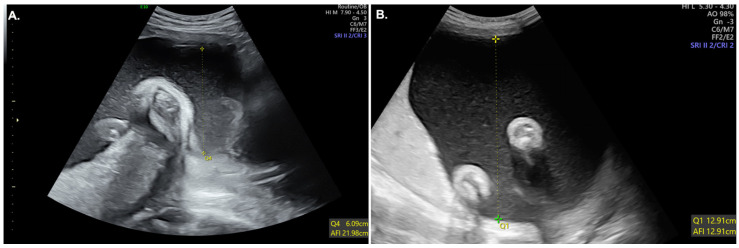
Changes in the amniotic fluid index (AFI). (**A**) Two weeks before admission, the AFI was 22, which was slightly increased. (**B**) At the time of admission, the AFI was 39, indicating polyhydramnios.

**Figure 2 medicina-59-00824-f002:**
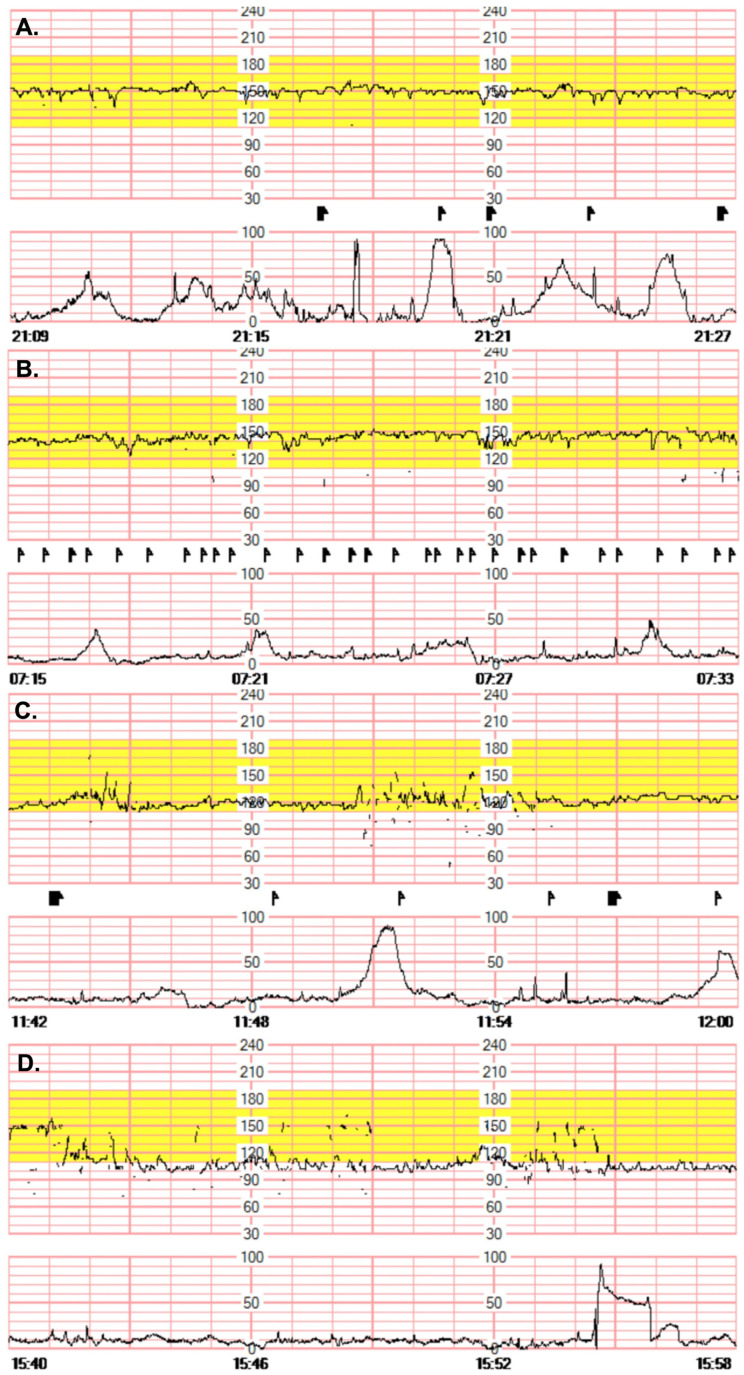
Changes in baseline fetal heart rate over time in a non-stress test (NST) (**A**) At admission, the baseline fetal heart rate was 150 bpm and showed reactive NST. Regular uterine contractions were observed at 2-min intervals. (**B**–**D**) NST on the day of delivery. The baseline fetal heart rate gradually decreased from 150 bpm to 100 bpm. An emergency cesarean section was performed due to persistent bradycardia.

## Data Availability

The data presented in this study are openly available.

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
