# Peer review of "Infection-Induced Rhabdomyolysis in a Pregnant Woman with Undiagnosed Myotonic Dystrophy: A Case Report"

_medicina, 2023, doi:10.3390/medicina59050824_

Round 1

Reviewer 1 Report

This manuscript is dealing with severe cases of rhabdomyolysis in pregnant women. 

The authors described a case of rhabdomyolysis caused by infection and full mutation of the myotonic dystrophy gene, which was not diagnosed before. Nevertheless, not all people who inherited autosomal dominant multi-systemic neuromuscular disorders -myotonic dystrophy with usual late adult onset manifestation know about it. Despite that, myotonic dystrophy type 1 is caused by a CTG expansion mutation in the 3’ UTR of dystrophia myotonica protein kinase (DMPK) in chromosome 19 q 13.3, with an estimated prevalence of 1:8000 ( in some European countries, even higher ) the broad spectrum of milder disease and late adult onset does not rise suspiciously to control everyone on this.

Unfortunately, this woman has been diagnosed getting a preterm newborn with full myotonic dystrophy manifestation. 

Nevertheless, GOOGLE ACADEMY has more than 20.000 publications on rhabdomyolysis topics and their etiology and pathophysiology. 

This case report does not give any novelty on this well-known topic.

Reviewer 2 Report

Dear Authors,

This a well-written and rare case presentation. As you concluded, myopathies may deteriorate during pregnancy.

1-Please allocate a paragraph in the discussion for myopathy and pregnancy complications.

2-Due to the genetic background of the disease discussion about the genetic issue (How to pass defective genes to children) is mandatory.

Round 2

Reviewer 1 Report

I agree with the authors that the case of myotonic dystrophy is a disease that requires more attention from specialists. This case report does not give any novelty on this well-known topic but encourages the clinical specialist to be suspicious of this possible manifestation.